# Residual Skill Policies:
# Learning an Adaptable Skill-based Action Space for Reinforcement Learning for Robotics

**Krishan Rana**[1], **Ming Xu**[1], **Brendan Tidd**[2], **Michael Milford**[1], **Niko Sünderhauf**[1]
[1]QUT Centre for Robotics, Queensland University of Technology
[2]Data61 Robotics and Autonomous Systems Group, CSIRO
`ranak@qut.edu.au`

**Abstract:**
Skill-based reinforcement learning (RL) has emerged as a promising strategy to leverage prior knowledge for accelerated robot learning. Skills are typically extracted from expert demonstrations and are embedded into a latent space from which they can be sampled as actions by a high-level RL agent. However, this *skill space* is expansive, and not all skills are relevant for a given robot state, making exploration difficult. Furthermore, the downstream RL agent is limited to learning structurally similar tasks to those used to construct the skill space. We firstly propose accelerating exploration in the skill space using state-conditioned generative models to directly bias the high-level agent towards only *sampling* skills relevant to a given state based on prior experience. Next, we propose a low-level residual policy for fine-grained *skill adaptation* enabling downstream RL agents to adapt to unseen task variations. Finally, we validate our approach across four challenging manipulation tasks that differ from those used to build the skill space, demonstrating our ability to learn across task variations while significantly accelerating exploration, outperforming prior works. Code and videos are available on our project website: `https://krishanrana.github.io/reskill`.

**Keywords:** Reinforcement Learning, Skill Learning, Transfer Learning

## 1 Introduction

Humans are remarkably efficient at learning new behaviours. We can attribute this to their ability to constantly draw on relevant prior experiences and adapt this knowledge to facilitate learning. Building on this observation, recent works have proposed various methods to incorporate the use of prior experience in deep reinforcement learning (RL) to accelerate robot learning.

One approach that has gained attention recently is the use of *skills* which are short sequences of single-step actions representing useful and task-agnostic behaviours extracted from datasets of expert demonstrations [1, 2, 3]. In the manipulation domain, for example, such skills could include *move-left*, *grasp-object* and *lift-object*. These skills are typically embedded into a latent space, forming the action space for a high-level RL policy. While yielding significant improvements over learning from *scratch*, there are still outstanding challenges for training RL agents using this latent *skill space*.

Firstly, naïve exploration by sampling randomly over all skills can be extremely inefficient [2, 3, 4]. Typical skill spaces encode an expansive set of generic skills, with only a small fraction of skills relevant for execution in a given robot state. Furthermore, the set of skills relevant to a given state do not typically cluster in the same neighbourhood in the skill space, i.e. the conditional density is *multi-modal*, making targeted sampling challenging. We address this issue of inefficient exploration by learning a *state-conditioned skill prior* over the skill space, which directly biases the agent's exploration towards sampling only those skills that are deemed relevant in its current state. Unlike previous works [3], our skill-prior captures the complex multi-modality of the skill space allowing us to sample directly from this density during exploration.

6th Conference on Robot Learning (CoRL 2022), Auckland, New Zealand.

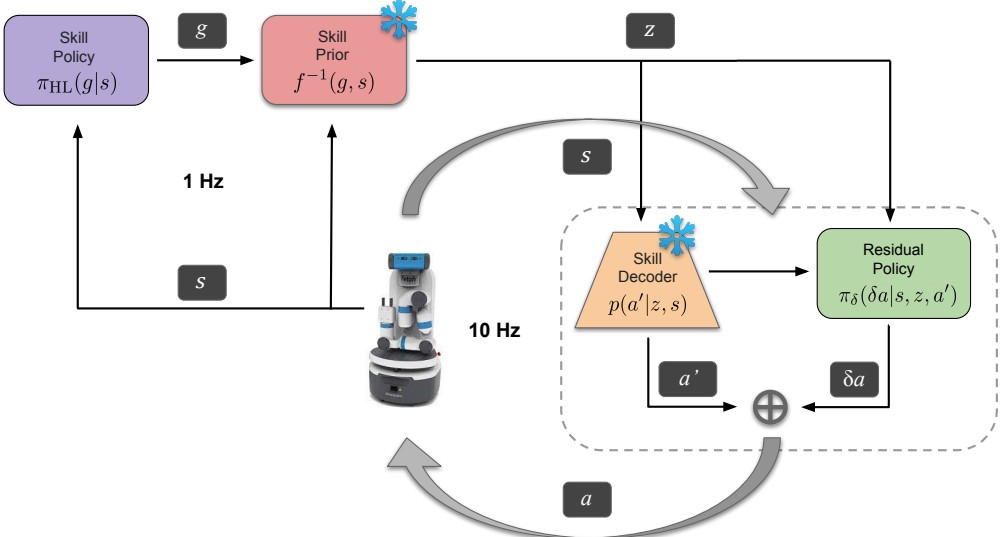

Figure 1: **Residual Skill Policies (ReSkill).** A skill-based learning framework for robotics. The skill prior transforms the agent's action space to a state-conditioned skill space using normalising flows, where only the relevant skills for the current state are explored. The residual policy allows for fine-grained adaptation of the skills to environment variations and unseen tasks. The ❄ symbol signifies that the pre-trained weights for these skill modules are frozen during downstream task learning.

Secondly, current approaches to skill-based RL assume that the skills are optimal and that downstream tasks are drawn from the same distribution used to create the skill space. Consequently, skill-based learning methods have limited generality and adaptability to task variations. For example, if the available skills were extracted solely from demonstrations of block manipulation on an empty table, learning a downstream task with obstacles, object variations, or different friction coefficients may not be possible. In this case, using a non-exhaustive skill space as an action space can cripple an RL agent's learning ability. We address this by introducing a low-level residual policy, which enables fine-grained *skill adaptation* of available skills to variations in the task not solvable by simply composing the available skills. This relaxes the need for exhaustive and expert demonstration datasets to one that can be extracted from existing classical controllers, which are a cheap and readily available source of demonstration data for robotics.

We summarise the main contributions of our approach denoted as Residual Skill Policies (ReSkill) as follows: **(1)** we propose a novel state-conditioned skill prior that enables direct sampling of relevant skills for guided exploration, **(2)** we introduce a low-level residual policy that can adapt the skills to variations in the downstream task, and **(3)** we demonstrate how our two contributions both accelerate skill-based learning and enable the agent to attain higher final performance in novel environments across four manipulation tasks, outperforming all baseline methods.

## 2 Related Work

**Skill-Based RL**    This has emerged as a promising approach to effectively leverage prior knowledge within the RL framework by embedding reusable skills into a continuous skill space via stochastic latent variable models [1, 2, 3, 5, 6]. Skill-based learning provides temporal abstraction that allows for effective exploration of the search space, with the ability to draw on previously used behaviours to accelerate learning. However as described previously, the skill space can be difficult to explore. Pertsch et al. [3] proposed the use of a prior over skills to help better guide exploration. They learn a unimodal Gaussian prior that approximates the density of relevant skills for a given state and regularise the downstream RL agent towards the prior during training to indirectly bias exploration. Singh et al. [4] proposed a more expressive strategy for learning these priors via the use of state-conditioned generative models. While this was conducted at the single-step action level, they show that they can directly bias exploration by sampling relevant actions from this density. In this work, we extend this idea to skill-based action spaces.

**Combining RL with Classical Control**    Our work is closely related to methods that seek to combine the strengths of classical control and RL for accelerated learning. In contrast to expert datasets or human demonstrators, classical control systems are cheap and readily available for most robotics tasks. Johannink et al. [7] proposed the residual reinforcement learning framework in which an RL agent learns an additive residual policy that modifies the behaviours of a predefined handcrafted controller to simplify exploration. Other works in this area have shown that combining the two modes of control allows for sample efficient initialisation [8, 9, 10], temporal abstraction for accelerated learning [11], and safe exploration [12]. While strongly motivating the re-use of existing handcrafted controllers, these methods are restricted to solving only one particular task for which the controller was designed, requiring a new controller for each new task. This limits the generality of these approaches to novel tasks. In this work, we decompose these controllers into task-agnostic skills and explore how they can be repurposed for learning a wide range of tasks.

**Hierarchical RL (HRL)**    These methods enable autonomous decomposition of challenging long-horizon decision-making tasks for efficient learning. Typical HRL approaches utilise a high-level policy that selects from a set of low-level temporally extended actions or options [13] that provide broader coverage of the exploration space when executed [14]. Prior work in hierarchical reinforcement learning can be divided into two main categories: learning low-level and high-level policies through active interaction with an environment [15, 16, 17, 18, 19], and learning options from demonstrations before re-composing them to perform long-horizon tasks through RL [20, 21, 22]. Our work shares similarities with both strategies. We follow the latter approach by pre-training a low-level skill module from demonstration data on which a high-level policy operates, while additionally learning a low-level residual policy online for fine-grained skill adaptation.

## 3    Problem Formulation

We focus on skill-based RL which leverages *skills* as the actions for a high-level policy $\pi_{\text{HL}}$, where a skill $\boldsymbol{a}$ is defined as a sequence of atomic actions $\{a_t, ..., a_{t+H-1}\}$ over a fixed horizon $H$. Skills are typically embedded into a continuous latent space $\mathcal{Z}$ from which $\pi_{\text{HL}}$ can sample during exploration [1, 2, 3]. Latent *skill spaces* expressively capture vast amounts of task-agnostic prior experience for re-use in RL. However, they pose several challenges that can negatively impact downstream learning. Firstly, current methods rely on expert and exhaustive demonstration datasets that capture skills for a wide range of potential task variations. Such datasets can be arduous and expensive to obtain. Secondly, as the skill space abstracts away the RL agent's atomic action space, the generality of skill-based RL is greatly limited to tasks that closely match the distribution of those captured by the dataset. This poses a trade-off between generality and sample efficiency in downstream learning [23]. Furthermore, latent skill spaces are expansive and can be arbitrarily structured, making random exploration difficult. Learning structure within this space can better inform exploration to accelerate downstream learning.

The downstream task is formulated as a Markov Decision Process (MDP) defined by a tuple $(\mathcal{S}, \mathcal{A}, \mathcal{T}, \mathcal{R}, \gamma)$ consisting of states, actions, transition probabilities, rewards and a discount factor. We learn the high-level policy $\pi_{\text{HL}}(z|s)$ to select the appropriate latent skills $z$ in a given state $s$ to maximise the discounted sum of $H$-step rewards $J = \mathbb{E}_{\pi_{\text{HL}}}[\sum_{t=0}^{T-1} \gamma^t \tilde{r}]$, where $T$ is the episode horizon, $\tilde{r}$ is the $H$-step reward defined as $\tilde{r} = \sum_{t=1}^{H} r_t$ and $r_t$ is the single-step reward.

## 4    Approach

We aim to re-purpose existing classical controllers as skills to facilitate RL in solving new tasks. We decompose demonstration trajectories produced by these controllers into task-agnostic skills and embed them into a continuous space $\mathcal{Z}$ using stochastic latent variable models. To facilitate exploration for downstream RL in this skill space, we propose a state-conditioned *skill prior* $p(z|s_0)$ over skills that directly biases the agent towards only sampling skills relevant to a given state based on prior experience. Finally, to enable truly general-purpose learning with the skill space, we propose a low-level residual policy $\pi_\delta(\delta a|\cdot)$ that can adapt the embedded skills to task variations not captured by the demonstration data. This gives the RL agent access to its single-step action space and relaxes the need for both expert and exhaustive datasets.

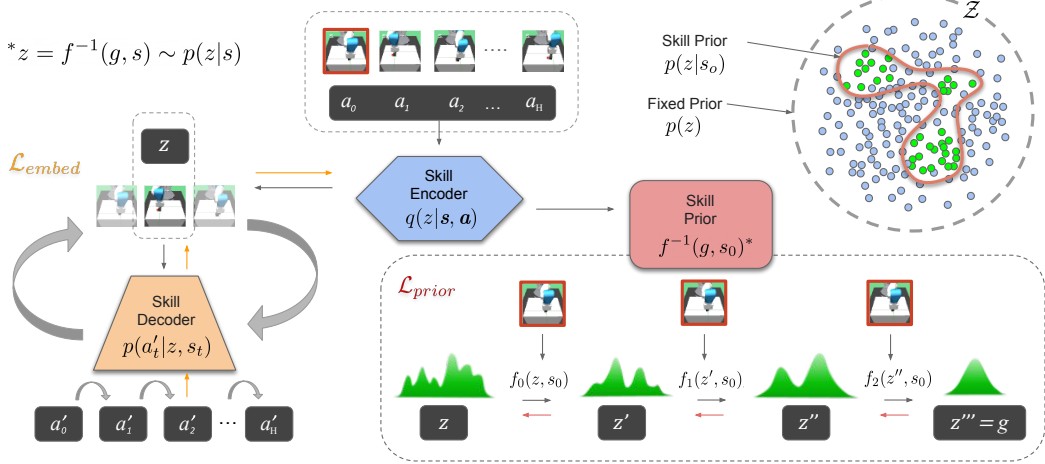

Figure 2: **Schematic for learning the state-conditioned skill space.** We train a VAE *embedding module* which encodes skills into a latent embedding space $\mathcal{Z}$. This module is comprised of an encoder and a closed-loop skills decoder, where the decoder recovers atomic actions from a latent skill embedding $z$. The *skill prior* module learns the state-conditioned density of useful skills based on the dataset. This conditional density is multi-modal in the skill space and we estimate it using normalising flows. Both modules are trained jointly, with coloured arrows illustrating the gradient flow between them.

Our approach can be decomposed into three sub-problems: (1) skill extraction from existing controllers, (2) learning a skill embedding and skill prior, and (3) training a hierarchical RL policy within this skill space with a low-level, residual skill adaptation policy. We discuss each of these steps in more detail below.

## 4.1 Data Collection

We re-purpose existing handcrafted controllers as sources of demonstration data for basic manipulation tasks involving *pushing* and *grasping* an object on an empty table. While these controllers are relatively simple, the trajectories produced possess a vast range of skills that can be recomposed to solve more complex tasks. We note here that these trajectories do not need to be optimal given our adaptable downstream RL formulation which we will describe in more detail in the following sections. The trajectories consist of state-action pairs and we perform unsupervised skill extraction by randomly slicing a $H$ length segment from them. We utilise both the extracted sequence of actions $\boldsymbol{a}$ as well as the corresponding states $\boldsymbol{s}$ to learn the state-conditioned skill space described in the next section. The state vector comprises of joint angles, joint velocities, gripper position and object positions in the scene, while the actions are continuous 4D vectors representing end-effector and gripper velocities in Cartesian space. More details about data collection and the specific handcrafted controllers used can be found in Appendix **??**.

## 4.2 Learning a State-Conditioned Skill Space for RL

Learning a state-conditioned skill space consists of two key steps: (1) embedding the extracted skills into a latent space; (2) learning a state-conditioned prior over the skills which we can sample from during exploration. Figure 2 provides a summary of our approach and we describe each component in detail below.

### 4.2.1 Embedding the skills

We follow the approach proposed in prior works [1, 24], where skills $\boldsymbol{a}$ are embedded into a latent space using a variational autoencoder (VAE) [25]. The VAE comprises of a probabilistic encoder $q_\phi$ and decoder $p_\theta$ with network weights $\phi$ and $\theta$ respectively, which maps a skill action sequence to

a latent embedding space and vice versa. Our encoder $q_\phi(z \mid \boldsymbol{s}, \boldsymbol{a})$ jointly processes the full state-action sequence, while our decoder $p_\theta(a_t \mid z, s_t)$ reconstructs individual atomic actions conditioned on current state $s_t$ and skill embedding $z$. The loss function for our VAE for a single training example is given by

$$\mathcal{L}_{embed} = \mathbb{E}_{q_\phi(z \mid \boldsymbol{s}, \boldsymbol{a})} \left[ \sum_{t=0}^{H-1} \log p_\theta\left(a_t \mid z, s_t\right) \right] - \beta D_{KL}(q_\phi\left(z \mid \boldsymbol{s}, \boldsymbol{a}\right) \parallel p(z)), \qquad (1)$$

where $p(z) \sim \mathcal{N}(0, I)$ and $\beta \in [0, 1]$ is the weighting for the regularisation term [25]. The *closed-loop* nature of the decoder considers the current robot state $s_t$ as well as sampled skill $z$ when recovering actions, which was found to significantly improve performance in downstream learning [26], particularly in dynamic environments.

### 4.2.2 Learning the state-conditioned skill prior

The nature of the VAE reconstruction loss described in Equation (1) allows us to sample skills from the latent space by using the prior $p(z) \sim \mathcal{N}(0, I)$. However, sampling directly from the prior can be inefficient, since the samples will range across the full, expansive set of available skills as well as regions that may not constitute any embeddings as shown in Figure 3. In reality, for a particular robot state, only a small subset of these skills are relevant for exploration. To address this inefficiency, we propose to learn a *state-conditioned skill prior* over the latent embedding space, which allows us to more frequently sample the *relevant skills* likely to be useful in a given state, thus accelerating exploration.

Mathematically, we are looking to learn a conditional probability density over the latent skill space $p(z|s_0)$ that we can draw samples from when training $\pi_{HL}$. This conditional density is typically highly multi-modal, i.e. for a given robot state, many skills can be relevant that may be far apart in the topology

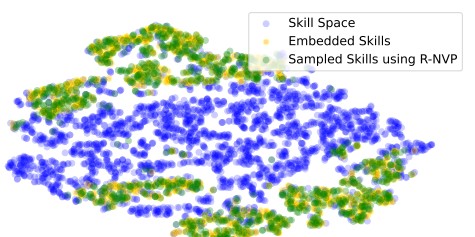

Figure 3: **t-SNE of the skill embedding space.** Blue denotes samples drawn from the VAE prior $p(z)$. Yellow denotes embedded skills drawn from the dataset. Green denotes samples using our trained skill prior. Note the multi-modality across the latent space. The skill prior samples from regions containing embedded and relevant skills, reducing the chances of sampling irrelevant regions (blue) not containing any meaningful skills.

of the latent skill space. Prior works [3] propose a simple unimodal density (e.g. Gaussian) which does not sufficiently capture this multi-modality and hence is unsuitable to directly sample skills from.

One of our contributions is around gracefully handling this multi-modality by using the real-valued non-volume preserving transformations (real NVP) method by Dinh et al. [27] based on normalising flows [28]. Conditional real NVP [29] learns a mapping $f : \mathcal{Z} \times \mathcal{S} \to \mathcal{G}$ which we can use to generate samples from $p(z|s)$. Spaces $\mathcal{Z}$ and $\mathcal{G}$ are identical, i.e. $\mathcal{Z} \cong \mathcal{G} \cong \mathbb{R}^d$ and furthermore $f$ is bijective between $\mathcal{Z}$ and $\mathcal{G}$ for a fixed state $s$. Sampling is achieved as follows: first sample $g \sim p_\mathcal{G}(g)$ from a simple distribution (e.g. $p_\mathcal{G}(g) \sim \mathcal{N}(0, I)$) and then transform $g$ into skill space $\mathcal{Z}$ using $f^{-1}$, i.e. $f^{-1}(g, s) \sim p(z|s)$. Since $f$ fully defines $p(z|s)$, we henceforth refer to $f$ as the skill prior.

We refer the reader to Dinh et al. [27] for more information about real NVP. We train our state-conditioned skill prior jointly with our VAE embedding module. Figure 3 illustrates the multi-modality present in the latent skill space and how our skill prior captures this. Implementation details are provided in Appendix **??**.

### 4.3 Reinforcement Learning in a State-Conditioned Skill Space

Once trained, the decoder and skill prior weights are frozen and incorporated within the RL framework. Our high-level RL policy denoted $\pi_{\text{HL}}(g|s)$ is a neural network that maps a state to a vector $g \in G$ in the domain of the skill prior transformation $f^{-1}$ defined in Section 4.2.2. The skill prior $f^{-1}$ is then used to transform $g$ to latent skill embedding $z$. Our use of normalising flows for our skills prior is motivated by [4], who hypothesised that the bijective nature of $f$ allows the RL agent

**Algorithm 1** Action Selection
***
**Given:** high-level policy $\pi_{\mathrm{HL}}$, residual policy $\pi_\delta$, skill decoder $p$, skill prior $f$
**Inputs:** state $s_t$
1:  $g \sim \pi_{\mathrm{HL}}(g|s_t)$                            ▷ sample a vector $g$ from the high-level policy
2:  $z \leftarrow f^{-1}(g, s_t)$                       ▷ map $g$ to a state-relevant skill using the skill prior
3:  **for** skill horizon $H$ **do**
4:      $a' \leftarrow p(a'|s_t, z)$          ▷ decode the skill to a single-step action using the skill decoder
5:      $\delta a \sim \pi_\delta(\delta a|s_t, z, a')$             ▷ sample a corrective action from the residual policy
6:      $a_t = a' + \delta a$                       ▷ sum the decoded and residual action
7:      **return** action $a_t$          ▷ execute the combined action in the environment and continue

to retain full control over the broader skill space $\mathcal{Z}$: for any given $z$, there exists a $g$ that generates $z = f^{-1}(g, s)$ in the original MDP. This means that all skills from $\mathcal{Z}$ can be sampled by the skill prior with non-zero probability.

Once a latent skill $z$ is attained by the high-level agent, the closed-loop decoder can reconstruct each action $a'$ sequentially conditioned on the current state for the skill horizon $H$. To increase the diversity of downstream tasks that can be learned using the fixed skill space, we introduce a low-level residual policy $\pi_\delta$ that can adapt the decoded skills at the executable action level, providing the hierarchical agent with full control over the MDP. We condition this residual policy on the current state $s$, selected skill $z$, and the action proposal generated by the skill decoder $a'$. The produced action $\delta a \sim \pi_\delta(\delta a|s, z, a')$ is added to the decoded action $a'$ before being executed on the robot. This low-level system is operated at a frequency of $H$ Hz based on the skill horizon, while the outer loop operates at 1 Hz. Figure 1 summarises our Residual Skill Policy (ReSkill) architecture and pseudo-code for action selection is provided in Algorithm 1. Given the non-stationarity imposed by the low-level residual policy being trained jointly with the high-level policy, we utilise an on-policy RL algorithm for training the hierarchical agent. Implementation details are provided in Appendix **??**.

## 5   Experiments

We evaluate our approach in a robotic manipulation domain simulated in MuJoCO [30], involving a 7-DoF Fetch robotic arm interacting with objects placed in front of it as shown in Figure 4. Evaluation is performed over four downstream tasks, each one not solvable purely with the prior controllers used for data collection as described in Appendix **??**. Each task is adapted from the residual reinforcement learning set of environments proposed by Silver *et al.* [31]. We describe each of the downstream tasks in detail in Appendix **??**.

We compare our approach against several state-of-the-art RL methods; these methods either operate directly on the single-step action space or train agents using a skills-based approach. The specific methods we compared against are as follows:

1. **Scripted Controller:** The average performance of the handcrafted controllers used for data collection across the 4 environments.

2. **Behavioural Cloning (BC) + Fine-Tuning:** Trains an agent by first using supervised learning on state-action pairs from the training set, followed by RL fine-tuning.

3. **SAC** and **PPO:** Trains an agent using Soft Actor-Critic [32] and Proximal Policy Optimisation [33] respectively with random weight initialisation.

4. **HAC:** Trains an agent using Hierarchical Actor-Critic [19] using a hierarchy of 2 for all tasks.

5. **PaRRot:** Leverages flow-based density estimation to map the agent's action space to single-step actions that are likely to be useful in a given state [4].

6. **SPiRL:** A hierarchical skill-based approach utilising KL regularisation of the RL policy towards a Gaussian skill prior [3]

7. **ReSkill (No Skill Prior):** Ablation study where the high-level policy samples skills directly using $\pi_{\mathrm{HL}}(z|s)$ with no skill prior guidance.

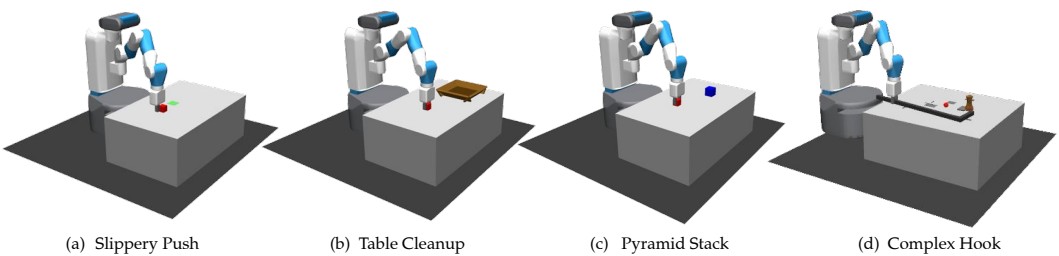

| (a) Slippery Push | (b) Table Cleanup | (c) Pyramid Stack | (d) Complex Hook |

Figure 4: **Downstream Tasks.** Robotic manipulation tasks involving a 7 DoF manipulator arm from Fetch Robotics. Each task exhibits a physical or dynamical variation to the environment that was not present in the skill extraction environment.

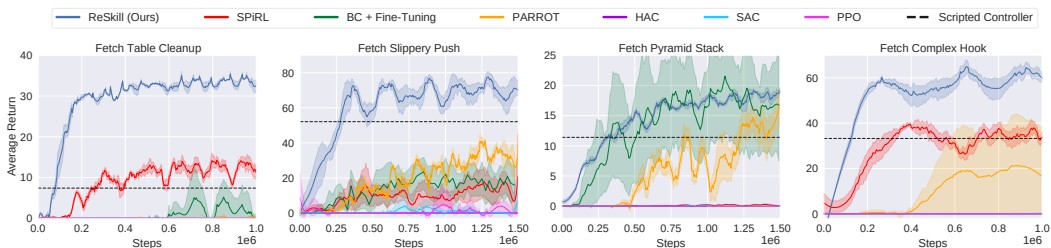

Figure 5: **Training Performance.** Average training performance across 5 random seeds for the different tasks. ReSkill outperforms all the baselines in both sample efficiency and convergence to the highest-performing final policy.

8. **ReSkill (No Residual):** Ablation study where we remove the low-level residual policy for adapting skills.

# 6   Results

The results across each task are summarised in Figure 5. Our approach, ReSkill, outperforms all the baselines, attaining the highest performing policy given its ability to adapt to task variations while additionally converging with superior sample efficiency given the guidance of the state-conditioned skill prior. Note, the baseline scripted controllers are sub-optimal across all four tasks as indicated by the dashed black line. SAC, PPO and HAC fail to learn altogether, given the inability of random exploration alone to yield any meaningful behaviours in sparse reward settings. Results are incrementally better using BC initialisation and fine-tuning; however, the agent still attains sub-optimal performance with high variance across the trials. A critical difference between PaRRot and ReSkill is the use of skills in ReSkill instead of single-step actions. The temporal abstraction provided by skills allows ReSkill to significantly outperform PaRRot in sample efficiency across all tasks. Further, we note that PaRRot struggles to learn a high-performing policy in the *Slippery-Push* and *Complex-Hook* domain which we could attribute to its inability to deal with the variations in the downstream task given the constrained action space. SPiRL accelerates learning, yielding better sample efficiency than all atomic-action-based methods on *Table-Cleanup* and *Complex-Hook*. However, it still fails to attain high final policy performance as it only composes high-level skills without the ability to adapt them to task variations, as shown by ReSkill.

We additionally provide an ablation study in Figure 6 to better understand the impact of the skill prior and residual agent on the overall skill-based learning architecture. ReSkill (No Skill Prior) demonstrates the importance of our sampling-based prior in accelerating learning and enabling our agent to learn altogether in the more complex stacking task. The ReSkill (No Residual) plots demonstrate the importance of the residual in enabling the agent to attain high final performance by re-introducing access to the single-step action space for fine-grained control. This allows the agent to appropriately balance the trade-off between generality and sample efficiency when leveraging skills in RL.

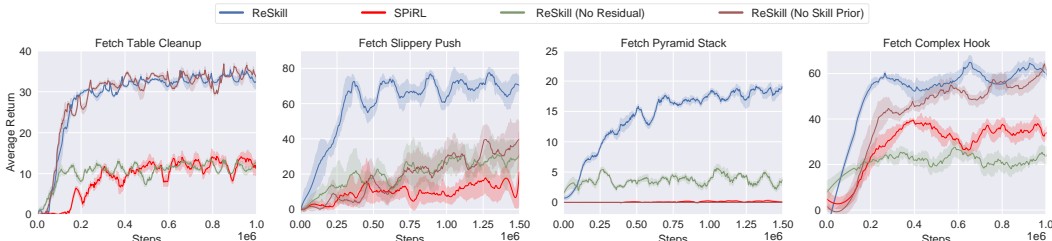

Figure 6: **Ablation Study.** Analysing the impact of the residual and skill prior on the average training performance across the 4 tasks. The skill prior plays an important role in accelerating learning with the more difficult tasks, while the residual is important for attaining higher rewards by adapting the skills to the variations in the training environment.

Furthermore, comparing ReSkill (No Residual) to SPiRL, we note that our approach attains faster convergence to the final policy. We can attribute this to the nature of our skill prior which enables the agent to directly sample and execute state-relevant skills from the early stages of training. In contrast, SPiRL relies on an indirect bias to the agent's exploratory actions via KL-regularisation towards a unimodal skill prior. This requires an initial "burn-in" phase before the policy is appropriately regularised towards the skill prior in order for the agent to start sampling relevant skills. We provide a more detailed ablation study of the impact of our proposed skill-prior in Appendix **??**.

### 6.1 Limitations

While we demonstrate significantly faster learning and better asymptotic performance than all the baseline methods, ReSkill still requires thousands of environment samples before converging to a high success rate. This limited our ability to demonstrate our method on a physical robot. An interesting future direction to explore would be to integrate our approach with an off-policy RL algorithm such as SAC in order to make better use of exploratory experience for improved sample efficiency. This, however, would require careful consideration of the non-stationarity imposed by the low-level residual component. Secondly, while we note that the skill prior used in this work is bijective and should theoretically allow the agent to access the entire range of skills in the larger skill space, the likelihood of selecting useful skills not seen in the training dataset becomes very low. This would particularly be an issue when there is a significant mismatch between the training dataset and the downstream task. We note here that the residual policy could play a major in addressing this, however, we leave a thorough evaluation of this to future work. Finally, learning the state-conditioned skill space requires two modules, specifically the VAE embedding module and flows-based skill prior module. Future work will explore using a single generative model such as a conditional VAE [34] to jointly handle the tasks of both modules.

## 7 Conclusion

This work proposes Residual Skill Policies (ReSkill), a general-purpose skill-based RL framework that enables effective skill re-use for adaptable and efficient learning. We learn a high-level skill agent within a state-conditioned skill space for accelerated skill composition, as well as a low-level residual agent that allows for fine-grained skill adaptation to variations in the task and training environment. Additionally, we show that our framework allows us to re-purpose a handful of scripted controllers to solve a wide range of unseen tasks. We evaluate ReSkill across a range of manipulation tasks and demonstrate its ability to learn faster and achieve substantially higher final returns across all task variations than the baseline methods. We see this as a promising step towards the effective re-use of prior knowledge for efficient learning without limiting the generality of the agent in learning a wide range of tasks.

**Acknowledgments**

K.R., M.X., M.M. and N.S. acknowledge the continued support from the Queensland University of Technology (QUT) through the Centre for Robotics. This research was additionally supported by the QUT Postgraduate Research Award. The authors would like to thank Robert Lee, Jad Abou-Chakra, Fangyi Zhang, Vibhavari Dasagi, Tom Coppin and Jordan Erskine for their valuable discussions towards this contribution. We would also like to thank the reviewers for their insightful and constructive comments towards the significant improvement of this paper.

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
