# OpenReview forum: "Residual Skill Policies: Learning an Adaptable Skill-based Action Space for Reinforcement Learning for Robotics"
_robot-learning.org/CoRL/2022/Conference — CoRL 2022 Poster_

### Official Review · Reviewer_fBFD · 2022-07-29

**Originality:** Good
**Technical Quality:** Good
**Clarity Of Presentation:** Very Good
**Impact:** 3

**Recommendation:**

Weak Accept: I recommend accepting the paper, but will not argue for my recommendation if the majority of other reviewers have a different opinion.

**Summary:**

This paper extends skill-prior RL (SPiRL) by (1) replacing a uni-modal gaussian skill prior with a multi-modal skill prior and (2) adapting low-level skills using a residual skill policy. The proposed multi-modal skill prior not only handles multi-modal skill distributions but also fully constrains exploration within the skill prior distribution. In addition, this paper addresses the problem of the imperfect skill repertoire by allowing low-level skill adaptation. The experiments demonstrate its faster convergence and higher performance over state-of-the-art RL and skill-based RL approaches.

**Issues:**

See the weaknesses listed above.

**Quality Of The Limitations Section:**

Limitations are addressed clearly

**Reviewer Expertise:**

4: The reviewer is confident but not absolutely certain that the evaluation is correct

**Robotics Focus:**

Highly relevant to robotics but no hardware experiments

**Strengths And Weaknesses:**

### Strenghts

* This paper well combines the existing techniques of multi-modal prior and residual policy learning to address the challenges in skill-based RL.

* The paper is clearly written.

### Weaknesses

* The tasks used for evaluation are too simple compared to the ones used in prior works. Testing the proposed method with more diverse skill sets would be necessary to prove the strength of the proposed skill prior.

* By utilizing normalizing flows to implement the multi-modal skill prior, the action space of the high-level policy becomes strictly limited to its skill prior distribution, which may not align well with the downstream task and is not flexible to acquire a new skill distribution. This method alleviates this issue by fine-tuning the low-level controller with the residual skill policy; however, this results in non-stationary training of the high-level policy, which is one of the major challenges in HRL.

* The limitation of the proposed method in the use of on-policy RL algorithms may hurt the sample efficiency of skill-based RL.

* The experiments are conducted in newly designed environments and tasks, which are tailored to the proposed method. Moreover, the tasks are rather simple and short-horizon compared to tasks where skill-based RL is good at. To demonstrate the broad applicability of the proposed method and better comparisons to prior works, it is recommended to perform experiments on commonly used benchmarks for skill-based RL, e.g., FrankaKitchen and PointMaze.

* Many implementation details, such as skill length H, task episode length, and task reward, are missing, which makes it hard to judge the correctness of the results.

### Questions and suggestions

* In Section 4.2.1, the proposed way of embedding skills to the latent skill space has been already proposed in Pertsch et al. [1] and demonstrated its superior performance.

[1] Pertsch et al., Demonstration-Guided Reinforcement Learning with Learned Skills, CoRL 2021

* In Figures 5 and 6, using more intuitive measures, such as success rate, would help understand the results. Or, at least, it should contain an explanation about the returns (rewards) for each task.

* In the references, many conference proceedings appear as arXiv preprints. Please use the proper citation.

* The appendix should be submitted separately. This paper violates the official guideline.

**Summary Of Recommendation:**

This paper proposes an intuitive and meaningful extension of SPiRL. However, the experiments are done in a new set of tasks, and many details are missing, which makes the results less convincing. Therefore, I stand for weak rejection.


**After rebuttal**

I appreciate detailed responses and additional experiments in the revised paper. The responses and additional experiments make this paper more solid, so I would raise my score to "Weak accept". But, I think this paper needs significant improvement to make it at the level that I can confidently accept this paper.

Here are additional comments for the revised paper:
* In Figure 8, ReSkill and ReSkill (No residual) should be reversed.
* L526-527, I do not agree with the claim that "substantially outperform SPiRL in both sample efficiency and convergence
527 to a higher final policy performance". Although ReSkill learns faster at the beginning, it could be due to better pre-training of skills and skill priors, and ReSkill and SPiRL seem to converge at not significantly different final performances.
* In addition, the last subtask is the only task that requires adaptation as the first three subtasks are present in the dataset. However, the proposed method seems to fail at adapting to this new subtask, which weakens the major claim of the paper.

---

> ### Author Response · Authors · 2022-08-27
> **Author Response**
>
> We thank the reviewer for their detailed and constructive feedback. We address each of the raised concerns across two parts and our detailed responses are provided below.

---

> > ### Author Response · Authors · 2022-08-27
> > **Part 2**
> >
> >
> > **The limitation of the proposed method in the use of on-policy RL algorithms may hurt the sample efficiency of skill-based RL.**
> >
> > While it is intuitive that using an on-policy algorithm could hurt the overall sample efficiency, this is not what we observed in our experiments: ReSkill surpassed the overall sample efficiency of SPiRL, which is an off-policy algorithm based on SAC. We agree that a combination of our strategy with off-policy methods could yield even better sample efficiency during training and have added a discussion around this point in the revised paper.
> >
> > **The experiments are conducted in newly designed environments and tasks, which are tailored to the proposed method. Moreover, the tasks are rather simple and short-horizon compared to tasks where skill-based RL is good at. To demonstrate the broad applicability of the proposed method and better comparisons to prior works, it is recommended to perform experiments on commonly used benchmarks for skill-based RL, e.g., FrankaKitchen and PointMaze.**
> >
> > As requested, we have included a new evaluation of our approach in the FrankaKitchen environment used in prior work in order to demonstrate the strengths and broad applicability of our proposed skill prior for accelerating learning. To further evaluate the strengths of our overall ReSkill approach on more diverse tasks, we have introduced a new ”Fetch Hook” environment that requires longer horizon planning, where an agent has to manipulate an object that is out of reach of its grippers. In order to solve the task the agent has to use a hook placed in the scene to reach the block and push it to a given goal. To add to the complexity of the downstream task, we introduce “bumps” on the table surface which make pushing and pulling the object more difficult, and additionally draw a random object at the start of each episode from a dataset of everyday objects each exhibiting different shapes, size and mass.
> > Noting the addition of new experiments in the revised paper, here we provide context for our original choice of experiments. The original Fetch manipulation tasks are from the residual RL literature [33] where each environment was designed to exhibit variations from a baseline environment in order to test the ability of a baseline pre-trained agent to adapt to these changes. This design choice makes them well suited to study the ability for our low-level residual policy to adapt skills to a wide range of task variations. This is an important ability to increase the generality of skill-based learning.
> > The environments used in previous skill RL literature focus primarily on skill recomposition and not skill adaptation, where the skill training dataset is typically drawn from the same distribution of tasks as those that the agent would be exposed to in downstream learning. In these contrived cases, the impact of the residual is minimal. We believe the new experiments we have added in these environments demonstrate the broad applicability of our method and its ability to compete in the skill-based RL domain in learning complex, long-horizon behaviours.
> > [33] Silver, K. Allen, J. Tenenbaum, and L. Kaelbling. Residual policy learning. arXiv preprint 376 arXiv:1812.06298 , 2018.
> >
> > **Many implementation details, such as skill length H, task episode length, and task reward, are missing, which makes it hard to judge the correctness of the results. In Figures 5 and 6, using more intuitive measures, such as success rate, would help understand the results. Or, at least, it should contain an explanation about the returns (rewards) for each task.**
> >
> > We have included the necessary implementation details of our approach in Appendix A and have added text to describe each task in more detail in Section 5 in order to provide better interpretability of the results illustrated in Figure 5 and 6.
> >
> > **In Section 4.2.1, the proposed way of embedding skills to the latent skill space has been already proposed in Pertsch et al. [1] and demonstrated its superior performance.**
> >
> > We thank the reviewer for bringing this to our attention and we have included the necessary citation of the work in the revised manuscript.
> >
> > **In the references, many conference proceedings appear as arXiv preprints. Please use the proper citation.**
> >
> > We apologise for this and have now updated all 11 citations in our revised manuscript to reflect the correct publishing entity.

---

> > ### Author Response · Authors · 2022-08-27
> > **Part 1**
> >
> > **The tasks used for evaluation are too simple compared to the ones used in prior works. Testing the proposed method with more diverse skill sets would be necessary to prove the strength of the proposed skill prior.**
> >
> > We note here that the tasks used in this work are drawn from the residual reinforcement learning literature, and were designed to demonstrate the ability for an existing agent to adapt to variations in its environment. This is well suited for the skill-residual setting we focus on in this work. To further demonstrate the strengths and broad applicability of our approach, we have included a new evaluation of ReSkill in the FrankaKitchen environment used in prior work. We have additionally introduced a new ”Fetch Hook” environment that requires longer horizon planning, where an agent has to manipulate an object that is out of reach of its grippers. In order to solve the task the agent has to use a hook placed in the scene to reach the block and push it to a given goal. To add to the complexity of the downstream task, we introduce “bumps” on the table surface which make pushing and pulling the object more difficult and additionally draw a random object at the start of each episode from a dataset of everyday objects each exhibiting different shapes, mass and friction. Across all these new tasks, we demonstrate that ReSkill can significantly outperform all baselines in both sample efficiency and final policy performance.
> >
> > **By utilizing normalizing flows to implement the multi-modal skill prior, the action space of the high-level policy becomes strictly limited to its skill prior distribution, which may not align well with the downstream task and is not flexible to acquire a new skill distribution. This method alleviates this issue by fine-tuning the low-level controller with the residual skill policy; however, this results in non-stationary training of the high-level policy, which is one of the major challenges in HRL.**
> >
> > The normalizing flow formulation is bijective and in theory, the agent can still sample all skills from the larger skill space with non-zero probability, however most of the probability mass will be placed on the relevant skills. We agree with the reviewer that this can be a limitation. However, we note that this problem specifically arises when there is a significant mismatch between the training dataset and the downstream task. We conducted a new additional empirical study to better understand how the skill prior impacts exploration and have included this in Appendix B.  We recorded the proportion of exploratory steps that resulted in block manipulation out of a total of 20k exploratory steps. Our skill prior directs at least 40% of the exploratory behaviours towards block manipulation allowing for a higher chance of executing reward yielding behaviours. While directing exploration towards the block, the agent is still able to explore alternative skills for potentially better solutions more suited to the task at hand. This empirical study shows that our skill prior formulation provides the agent with the ideal balance of guided exploration for accelerated learning without over constraining it from identifying alternative strategies. Re: “This method alleviates this issue by fine-tuning the low-level controller with the residual skill policy” - we clarify here that this is not the intention of the residual in ReSkill, but we acknowledge that this is a possible outcome of training if the skill prior is too constrained. The main purpose of the residual in our case is to adapt existing and relevant skills selected by the skill prior to better suit the task at hand in order to yield higher performance.

---

### Official Review · Reviewer_daKY · 2022-08-01

**Originality:** Very Good
**Technical Quality:** Excellent
**Clarity Of Presentation:** Excellent
**Impact:** 3

**Recommendation:**

Weak Accept: I recommend accepting the paper, but will not argue for my recommendation if the majority of other reviewers have a different opinion.

**Summary:**

This work introduces ReSkill, a method for skill-based RL that enables efficient exploration and adaptation to tasks beyond the initial set of skills. The method is composed of three main parts that are trained sequentially: skill extraction from scripted policy demonstrations, learning a skill embedding and state-conditioned skill prior, and learning a hierarchical policy that combines an action decoder for sampled skills with a learned residual policy. The two main competitive baselines that the paper compares against are PaRRot (non-hierarchical version of ReSkill that uses a similar flow-based state-conditioned generative model for direct action prediction instead of skill prediction) and SPiRL (this can be viewed as ReSkill without the state-conditioned skill prior). ReSkill outperforms all baselines, and ablations show that the skill prior is useful for exploration and the residual policy is useful for adapting to changed environments.

**Issues:**

- The test environments differ quite a bit from the training environment used to generate the skills (push and pick and place). As tasks increase in complexity and the test skills are increasingly different from the training skills, the residual policy becomes very important. Therefore, how does ReSkill scale to high-dimensional environments or very diverse tasks, especially as there is an increasing distribution shift between train and test skills?
- Why is “ReSkill (No Skill Prior)” in Figure 6(a) so much better than SPiRL? Shouldn’t they be very similar?
- Is the most novel / important contribution the state-conditioned skill prior or the residual policy, given that they both seem important for performance?


**Quality Of The Limitations Section:**

Limitations are addressed clearly

**Reviewer Expertise:**

4: The reviewer is confident but not absolutely certain that the evaluation is correct

**Robotics Focus:**

Relevant but unlikely to deploy to hardware in near future

**Strengths And Weaknesses:**

Strengths:
- The method’s state-conditioned skill prior is a powerful abstraction and a very clear improvement over the prior work SPiRL in terms of exploration benefit and skill space expressivity.
- The paper is well presented with great clarity. Each of the method’s three components are explained concisely. The system design and motivations are visualized clearly in Figures 1, 2, and 3.

Weaknesses:
- The method is only evaluated on simple simulated domains; it is unclear how the method scales with higher dimensional tasks (with complex, high-dimensional skills) where state-conditioning may be intractable


**Summary Of Recommendation:**

The paper is presented extremely well and clearly describes the three main components of their method with ample motivation and analysis. The comparisons to relevant baselines and ablations showcase the effectiveness for exploration of the main contribution of the paper, the state-conditioned skill prior via density estimation. While the method is very compelling and technically sound, the main reservation I have is that the method is only evaluated on relatively simple domains, so it is unclear how it extends to more complex domains and especially the real world.

---

> ### Author Response · Authors · 2022-08-27
> **Author Response**
>
> We thank the reviewer for their detailed and constructive feedback. We address each of the raised concerns across two parts and our detailed responses are provided below.

---

> > ### Author Response · Authors · 2022-08-27
> > **Part 2**
> >
> > **Why is “ReSkill (No Skill Prior)” in Figure 6(a) so much better than SPiRL? Shouldn’t they be very similar?**
> >
> > ReSkill (No Skill Prior) is capable of attaining a much higher performance than SPiRL, because ReSkill has a low-level residual policy that can adapt the available skills to the changes in friction seen in the environment, allowing the agent to reach high rewards. SPiRL is limited to its set of fixed skills which are not suitable for use in this downstream environment. Furthermore we believe the closed-loop nature of ReSkill’s skill decoder provides the agent with better control of the block, in contrast to the open-loop decoder used in SPiRL.
> >
> > **Is the most novel / important contribution the state-conditioned skill prior or the residual policy, given that they both seem important for performance?**
> >
> > The ReSkill architecture entails two key contributions to address current limitations in the skill-based RL literature and both the state-conditioned skill prior and residual policy play a role in addressing these limitations. We provide a new ablation study in Appendix B to support the effectiveness of our skill prior in guiding exploration towards meaningful regions of the skill-space. We additionally include a new thorough evaluation of the ReSkill (No Residual) and ReSkill (No Skill Prior) ablation to further support the necessity of each system for addressing the sample inefficiency and adaptability of skill-based learning.

---

> > ### Author Response · Authors · 2022-08-27
> > **Part 1**
> >
> > **The test environments differ quite a bit from the training environment used to generate the skills (push and pick and place). As tasks increase in complexity and the test skills are increasingly different from the training skills, the residual policy becomes very important. Therefore, how does ReSkill scale to high-dimensional environments or very diverse tasks, especially as there is an increasing distribution shift between train and test skills?**
> >
> > **The method is only evaluated on simple simulated domains; it is unclear how the method scales with higher dimensional tasks (with complex, high-dimensional skills) where state-conditioning may be intractable**
> >
> > We thank the reviewer for the specific feedback around the appropriateness of our experiments. We have added two new tasks which address the concerns around evaluation on more complex, higher-dimensional tasks directly.
> > The first task we have added is a more complex, long horizon manipulation task called “Fetch Hook”, originally proposed by [33] with a diverse range of variations from the collected skills to further evaluate our proposed system. This task requires the agent to manipulate an object that is out of reach from the agent’s grippers, requiring it to use a hook that is placed on the table. The downstream task exhibits 2 significant variations from the skill extraction environment: 1) the environment simulates “bumps” on the table surface which makes it difficult to push or pull the object towards the goal location and 2) the environment draws random objects from a dataset of “thingiverse” objects that exhibit a range of sizes, mass and friction. In order to be successful the agent has to be able to manipulate the range of objects. These are substantial variations from the baseline environment used for skill extraction and will require the downstream agent to both recompose the skills appropriately and adapt them at the low-level using the residual policy in order to successfully solve the task.
> > The second new task we have added to the experiments  is set in the FrankaKitchen environment, commonly used to evaluate skills-based RL methods, which requires the agent to manipulate a series of objects in the scene in order to successfully complete the task. The state space of the FrankaKitchen environment is 60-dimensional, which is substantially higher than the 19-dimensional task present in the existing environments.
> > In both new tasks, we demonstrate the ability of ReSkill to both accelerate learning and attain higher returns than all prior methods.
> > [33] . Silver, K. Allen, J. Tenenbaum, and L. Kaelbling. Residual policy learning. arXiv preprint 376 arXiv:1812.06298 , 2018.
> >
> > **While the method is very compelling and technically sound, the main reservation I have is that the method is only evaluated on relatively simple domains, so it is unclear how it extends to more complex domains and especially the real world.**
> >
> > We believe this point is addressed by our significant new experiments, as detailed in previous comments. We have evaluated our system on two new, substantially more complex tasks (Fetch Hook task) and domains (FrankaKitchen). Given that ReSkill has shown very promising results in simulation, we plan to deploy ReSkill on a real robot in future work. A major challenge and important consideration for real world deployment is that despite the significant progress made in the space of skill-based learning and the ability for ReSkill to surpass all prior methods in sample efficiency, the agent still requires upwards of 200k steps before it can converge to a meaningful policy. This is still an ongoing research area and we believe with additional modifications to our method including the use of an off-policy variant for the underlying RL algorithm, we could significantly reduce the number of samples required to train the agent in order to efficiently train it in the real world.

---

### Official Review · Reviewer_eiyG · 2022-08-02

**Originality:** Good
**Technical Quality:** Good
**Clarity Of Presentation:** Fair
**Impact:** 3

**Recommendation:**

Weak Accept: I recommend accepting the paper, but will not argue for my recommendation if the majority of other reviewers have a different opinion.

**Summary:**

This work provides two main improvements to robot skill learning methods: First is a latent skill space, which can be used as a prior for accelerating the learning on both the high-level policy $\pi_{HL}$ and the skill decoder. The second is a residual skill mechanism, which can further adjust the action generated by the skill decoder on different task settings.

The comparison of simulated environments shows the effectiveness of the proposed method and illustrated the advantage of learning with skill abstractions and pre-captured data.

**Issues:**

1. For experiments, why is the average reward of ReSkill (No Residual) different from ReSkill within the first 20k steps? As shown in the line.400, the residual policy is inactive so ReSkill (No Residual) and ReSkill will be exactly the same.
2. The analysis of ReSkill (No Skill Prior) is missing on Sec.6, and the experiment setting that uses samples from Z is ambiguous, does $z~N(0, I)$ or $z~\pi(z|s)&, maybe the latter would be better?
3. How is the high-level policy trained? can it be fine-tuned from prior $p(z|s)$? Is there any quantitative analysis on how much could the proposed prior narrows the exploration range?
4. More comparisons on Hierarchical RL methods are welcomed.

**Quality Of The Limitations Section:**

Additional details required

**Reviewer Expertise:**

3: The reviewer is fairly confident that the evaluation is correct

**Robotics Focus:**

Highly relevant to robotics but no hardware experiments

**Strengths And Weaknesses:**


This work introduces a latent skill prior, which can be used for accelerating both high-level policy learning and skill decoder optimization. When learning the high-level policy, the proposed latent skill embedding can reduce the variance when doing explorations and can be estimated with variational inferences as proposed by this work. Moreover, when learning the low-level policy (the skill model), a residual policy model is proposed to better fit the task requirements. These two ways of using pre-captured data are intuitive and appropriate.

However, this work has obvious shortcomings. Firstly, the paper is not so well organized and a little bit hard to read, some of the details are missed like how is the high-level policy defined and trained (why not directly use $\pi_{HL}=q(z|s)$ and fine-tuning it on tasks), and how will the skill horizon affect the learning performance and efficiency. Then, the survey on hierarchical RL is not sufficient, there exists some work that uses VAE/cVAE for skill discovery and policy learning, like [1,2] and so on, although they come from option-based hierarchical imitation learning, a certain comparative survey is necessary. Further, the comparison is somehow not sufficient, the comparison among HRL methods like Option AC[3], H-DDPG, and so on are welcome.

[1] Sharma M, Sharma A, Rhinehart N, et al. Directed-Info GAIL: Learning Hierarchical Policies from Unsegmented Demonstrations using Directed Information[C]//International Conference on Learning Representations. 2018.
[2] Hausman K, Chebotar Y, Schaal S, et al. Multi-modal imitation learning from unstructured demonstrations using generative adversarial nets[J]. Advances in neural information processing systems, 2017, 30.
[3] Bacon P L, Harb J, Precup D. The option-critic architecture[C]//Proceedings of the AAAI Conference on Artificial Intelligence. 2017, 31(1).

**Summary Of Recommendation:**

This work enhances the skill-based hierarchical RL method from two aspects: a latent prior that can be used for faster exploration and the residual low-level policy, where the technical innovation is limited. Also, the comparisons with HRL methods are insufficient and affect persuasiveness the of the results. Finally, the organization of this work is a little bit unclear, and the experiment and theory do not demonstrate the advantage of using latent prior (see Q.2, Q.3 below).

---

> ### Author Response · Authors · 2022-08-27
> **Author Response**
>
> We thank the reviewer for their detailed and constructive feedback. We address each of the raised concerns across two parts and our detailed responses are provided below.

---

> > ### Author Response · Authors · 2022-08-27
> > **Part 2**
> >
> > **Is there any quantitative analysis on how much could the proposed prior narrows the exploration range?**
> >
> > We have added a new ablation study in Appendix B to better analyse how the proposed prior narrows the exploration range. Firstly we have added Figure 7 which provides a visual depiction of the exploratory trajectories taken by 4 different methods used in prior literature. We show that our state-conditioned skill prior enables direct sampling of relevant skills from the skill space that are useful for solving the task at hand - object manipulation. For a more quantitative measure of how our skill prior impacts exploration, we additionally recorded the proportion of exploratory steps that resulted in block manipulation out of a total of 20k exploratory steps. Our skill prior directs at least 45% of the exploratory behaviours towards block manipulation for a higher chance of executing reward yielding behaviours, while still enabling the agent to explore alternative skills for potentially better solutions more suited to the task at hand. This empirical study shows that our skill prior formulation provides the agent with an ideal balance of guided exploration for accelerated learning without over constraining it from identifying alternative strategies. In contrast the comparative baselines attained a much lower proportion of useful behvaiours (<10%) which leads to their slower learning or inability to learn the task at hand as shown in Figure 6.
> >
> > **More comparisons on Hierarchical RL methods are welcomed.**
> >
> > We have added a new comparative HRL baseline across all tasks. In particular we utilise Heirarchical Actor Critic (HAC) requested by Reviewer 1.  While HAC does have the potential to work well on goal conditioned long horizon tasks, it is reliant on the agent occasionally reaching sub-goals that are close to the intended goal. With increasing task complexity, this requirement becomes increasingly unattainable when relying on random exploration: our experiments (Fig. 5 and Section 6) show that HAC is not able to learn the evaluated tasks. We provide an additional ablation study in Appendix B, which illustrates the limitations of the exploration strategy used by HAC. For future work, a combination of hierarchical approaches with our prior-guided exploration could yield agents that learn even faster than with the skill prior alone. We additionally include a more detailed related work discussion of HRL methods in the revised manuscript.
> >
> > **The paper is not so well organized and a little bit hard to read, some of the details are missed like how is the high-level policy defined and trained (why not directly use πHL=q(z|s) and fine-tuning it on tasks), and how will the skill horizon affect the learning performance and efficiency.**
> >
> > We have made substantial modifications and additions to the text, figures and other paper components to improve clarity and completeness, as detailed throughout the other responses. Here, we clarify that the high level policy is responsible for skill recomposition. The low-level policy is responsible for skill adaptation by fine-tuning these composed skills to variations in the downstream task. Both the high level and low level residual policy are trained simultaneously using PPO. We found that by training in this manner, we could make effective use of the online exploratory data collected to update both the high level and low level policy components, allowing for better overall sample efficiency. While the observation that this could be done in a two step process (i.e. first learning the high level component then fine tuning with the low level policy) we note that this strategy requires separate exploration phases for learning each system requiring a larger amount of online experience as opposed to jointly training the two systems. We detail the training of each RL policy component in Appendix A.0.2. We acknowledge that the skill horizon hyperparameter does have an impact on learning performance and efficiency and that this has been thoroughly evaluated in prior skill-based RL work [3].

---

> > ### Author Response · Authors · 2022-08-27
> > **Part 1**
> >
> > **The experiment and theory do not demonstrate the advantage of using latent prior (see Q.2, Q.3 below).**
> >
> > In Appendix B we describe a new experiment that demonstrates the positive effect of the skill prior on the agent’s exploration behaviour. When exploration is guided by  the skill prior, over 45% of actions result in meaningful behaviour, i.e. a manipulation of the environment. Other methods only achieve around 9% (skill space without prior) or even 0.5% (random exploration). This new empirical study shows that our skill prior formulation provides the agent with the perfect balance of guided exploration for accelerated learning without over constraining it from identifying alternative strategies.
> > The new Figure 7 illustrates the differences in exploration behaviour, again highlighting how our state-conditioned skill prior guides exploration towards meaningful actions, which results in accelerated learning.
> > Furthermore, we conducted new experiments across 2 additional environments (FrankaKitchen and Fetch Complex Hook). Both exhibit complex long horizon tasks. For the Complex Hook task, the skill prior plays an important role in accelerating learning when compared to both the ReSKill (no prior) variant and SPiRL (see Fig. 6). Without the skill prior, the agent completely fails to learn in the stacking task, further demonstrating the importance of this prior when learning (Fig. 6). These findings are consistent with the results achieved on the FrankeKitchen task: ReSkill with our proposed skill prior learns faster compared to the skill prior used by SPiRL.
> >
> > **For experiments, why is the average reward of ReSkill (No Residual) different from ReSkill within the first 20k steps? As shown in the line.400, the residual policy is inactive so ReSkill (No Residual) and ReSkill will be exactly the same.**
> >
> > We apologise for the confusion, and realised that we incorrectly mentioned that ReSkill does not use the residual in the first 20k steps, which we have corrected in updated text. In reality, we gradually introduce the residual from the start of training, allowing it to take full effect after 20k steps. We do this by weighting the additive residual component using a logistic function that gradually increases from 0 to 1 over the course of the first 20k steps. We found that this gradual introduction of the residual policy allowed for stable training of the hierarchical agent as opposed to a hard introduction at 20k steps. This explains the discrepancy between the average reward of ReSkill (No Residual) and ReSkill in the first 20k steps as the skills decoded by the prior are impacted by the randomness of the weighted residual policy actions. We have now clarified this in the implementation details section in Appendix A.0.2.
> >
> > **The analysis of ReSkill (No Skill Prior) is missing on Sec.6, and the experiment setting that uses samples from Z is ambiguous, does $zN(0, I) Or z\pi(z|s)&, maybe the latter would be better?**
> >
> > We have now added a new analysis of ReSkill (No Skill Prior) to the discussion section. We agree that the description for ReSkill (No Skill Prior) was ambiguous: we have now clarified this description in the manuscript as the training of an RL agent that directly samples from the high level policy $\pi_{HL}(z|s)$ being trained.

---

### Official Review · Reviewer_HjV5 · 2022-08-05

**Originality:** Good
**Technical Quality:** Very Good
**Clarity Of Presentation:** Very Good
**Impact:** 3

**Recommendation:**

Weak Reject: I recommend rejecting the paper, but will not argue for my recommendation if the majority of other reviewers have a different opinion.

**Summary:**

The paper addresses the problem of learning to execute and adapt existing skills to more complex tasks that are different from the tasks used to build the atomic skills. In particular,  (i) the paper collects state-action pairs of handcrafted controllers, (ii) learns a skill embedding and a state-conditioned skill prior based on this data and (iii) learns an RL policy that acts within this skill space and adapts each skill to task variations.

**Issues:**

Please address the weaknesses in the "Strengths And Weaknesses" section.

**Quality Of The Limitations Section:**

Limitations are addressed clearly

**Reviewer Expertise:**

3: The reviewer is fairly confident that the evaluation is correct

**Robotics Focus:**

Highly relevant to robotics but no hardware experiments

**Strengths And Weaknesses:**

# Strengths
- The paper is clearly and well written
- The idea to condition the skill prior on the current state seems clever, since it rules out state spaces for which the skill is unlikely to be helpful while the skill prior captures the fact that the same state might be part of multiple skills by the multimodal distribution.

# Weaknesses
- Evaluations only in simulation: data generated in reality might suffer from distributional shift and other problematic influences. It is still to be shown if the techniques presented in this paper can handle such difficulties. Can you discuss, for instance, how the skill embedding space in Figure 3 would change?
- It appears that the authors composed a new method out of several existing approaches. For instance, the autoencoder used for the skill embedding comes from [1,21]. Please make it very explicit what you're adding. Is it the multimodal structure of the skill prior? A contribution could also be the clever composition of these existing methods to reach higher empirical performance.
- It would be interesting to also compare to hierarchical RL approaches such as HAC [1]. It seems obvious that neither PPO nor SAC perform well in such tasks.
- Only three random seeds have been used to test the data. Since the progress in RL is generally very noisy – especially in more complex tasks – it is important to evaluate more thoroughly.

[1] Andrew Levy, George Dimitri Konidaris, Robert Platt Jr., and Kate Saenko. 2019. Learning multi-level hierarchies with hindsight. In Proceedings of the 7th International Conference on Learning Representations

**Summary Of Recommendation:**

Although the paper presents an interesting approach and addresses an imoortant research question, it also has multiple weaknesses. I encourage the authors to clarify and address these weaknesses. I would be happy to increase the score then.

---

> ### Author Response · Authors · 2022-08-27
> **Author Response**
>
> We thank the reviewer for their detailed and constructive feedback. We address each of the raised concerns across two parts and our detailed responses are provided below.

---

> > ### Author Response · Authors · 2022-08-27
> > **Part 2**
> >
> > **Evaluations only in simulation: data generated in reality might suffer from distributional shift and other problematic influences. It is still to be shown if the techniques presented in this paper can handle such difficulties. Can you discuss, for instance, how the skill embedding space in Figure 3 would change?**
> >
> > Thank you for raising this point; we acknowledge that a limitation of our evaluation is that it is performed entirely in simulation. We note we have now added new, substantially more difficult simulation experiments that further demonstrate the capability of the system to work in complex, challenging conditions. With regard to the distributional shift: this would relate in substantial part to the sim-to-real topic, which we are not addressing explicitly in this paper. For real-world deployment, which is the subject of future work, we would prefer to collect skills using real robot trajectories and also train the downstream RL agent on the robot itself, bypassing the distributional shift issue entirely. This approach would however require upwards of 20K demonstration trajectories in order to train the skill prior, an expensive exercise to collect on a real robotic system.
> > **Can you discuss, for instance, how the skill embedding space in Figure 3 would change?** We will comment on the sim-to-real gap in the context of our experiments specifically. Recall that we set the agent state to be directly related to the kinematic state of the robot as well as the objects in the environment. Not having to learn a policy from raw images (perception is handled separately) reduces the sim-to-real gap substantially. Assuming our simulated robot is kinematically identical to the real robot, we expect the skill space to be comparable (in terms of high-level behaviour) between simulation and reality. For instance, if the robot gripper is near the block, we would expect that, as in simulation, the real robot will also likely select skills that move towards the block for a push or grasp.

---

> > ### Author Response · Authors · 2022-08-27
> > **Part 1**
> >
> > **It appears that the authors composed a new method out of several existing approaches. For instance, the autoencoder used for the skill embedding comes from [1,21]. Please make it very explicit what you're adding. Is it the multimodal structure of the skill prior? A contribution could also be the clever composition of these existing methods to reach higher empirical performance.**
> >
> > Our main contributions are the introduction of a new state-conditioned skill prior, and the combination of skill-based RL with a low-level residual policy. These two contributions address a number of major shortcomings of existing skill-based approaches. Specifically, given that the skill space can be expansive and encodes a diverse range of behaviours, our state-conditioned skill prior enables us to capture the multimodality of the relevant embedded skills in a given state, from which the downstream RL agent can directly sample skills from. We show that this significantly accelerates learning by guiding exploration through the relevant regions of the skill space. We provide an additional ablation study to support this claim in Appendix B. The reviewer is correct in that prior works have previously proposed learning a skill embedding using a VAE, as we have noted in Section 4.2.1. However, these methods lacked the ability to guide exploration in the skill space, an important new capability which we address in this work using our state-conditioned skill prior described above. Furthermore, the skill-based methods proposed in prior work are highly dependent on the availability of the required skills for downstream tasks, and struggle to learn when these tasks exhibit variations for which the necessary skills are unsuitable. This limits the overall generality of the skill-based framework to novel environments. Here we address this by introducing a low level-residual policy that can appropriately adapt the available skills to task variations. This approach enables us to achieve higher empirical performance with regards to task returns and broader applicability of skill-based RL to a range of downstream tasks. The combination of these two systems leads to our approach outperforming all baselines, in both sample efficiency and final policy performance. We have made our specific contributions as described above clearer in the revised paper.
> >
> > **It would be interesting to also compare to hierarchical RL approaches such as HAC [1]. It seems obvious that neither PPO nor SAC perform well in such tasks.**
> >
> > We agree that HAC has the potential to work well on goal conditioned long horizon tasks and have now added HAC as an additional baseline to our experiments. It is important to note however, that HAC relies heavily on the agent occasionally reaching sub-goals that are close to the intended goal. With increasing task complexity, this requirement becomes increasingly unattainable when relying on random exploration: our experiments (Fig. 5 and Section 6) show that HAC is not able to learn the evaluated tasks. We provide an additional ablation study in Appendix B, which illustrates the limitations of the exploration strategy used by HAC. For future work, a combination of hierarchical approaches with our prior-guided exploration could yield agents that learn even faster than with the skill prior alone. We thank the reviewer for this interesting idea.
> >
> > **Only three random seeds have been used to test the data. Since the progress in RL is generally very noisy – especially in more complex tasks – it is important to evaluate more thoroughly.**
> >
> > Following the reviewer’s suggestion, we now report all experiments with five seeds, and have updated Figures 5 and 6 accordingly. ReSkill consistently shows low levels of variability across all seeds. ReSkill also consistently performs best overall, conclusively demonstrating the benefit of our proposed approach. While we can run even more experiments by increasing the seeds from 5 to 10 after the rebuttal period, based on the relationships and trends observed in moving from 3 to 5 seeds in our newly presented results, we do not expect to observe any significant change in performance or variation.

---

### Meta-Review · Area_Chair_s6zm · 2022-08-11

**Recommendation:** Accept (Poster)
**Confidence:** 3

**Metareview:**

AE summarizes the strength and weakness of the paper raised by reviewers as follows:
Strengths

- The paper is well-written

- The learning of the state-conditioned skill prior is interesting, and the implementation based on the normalizing flow appears reasonable

Weakness

- The experimental results could be improved by using more tasks and more random seeds

- The comparison with HRL methods is missing

The paper is well-written, and the proposed method appears to be reasonable. Meanwhile, the weakness of the experimental results seems the bottleneck. In the initial submission, results are reported based on new tasks, which seem tailored to the proposed method, and the scores are computed using only three random seeds.

=== post-rebuttal comments ===

Although reviewers raised several concerns in the initial review, most of concerns were resolved by additional experimental results provided by the authors. Thus, AE recommends the acceptance of the paper. However, as the amout of the revised texts are large, AE stroungly encourage the authors to go through the paper once again and carefully check the writing and presentation. Also, as Reviewer HjV5 indicated, it is necessary to discuss how the proposed method would be applied to a real robot. AE would like the authors to make sure to address these points before submitting the final version.



**Best Paper Nomination:**

No

---

> ### Author Response · Authors · 2022-08-27
> **Author Response**
>
> We thank the reviewers for their constructive and specific feedback on our work, which has enabled us to fully and substantially address their concerns in the revised manuscript. To address the feedback, we have substantially expanded the experiments and provided significant additional analysis in the paper. Key new additions and changes include: two new extra long horizon manipulation tasks; an additional HRL baseline for comparison; complete re-running of all experiments with an increased number of five seeds; and a new ablation study that better clarifies the substantial positive impact of the proposed skill prior during exploration. In this response document, we have highlighted major updates in blue in the uploaded PDF file. We now provide a detailed response to each and every of the reviewers’ concerns in the comments below.